# Quantitative lipidomic analysis of *Ascaris suum*

Tao Wang[1], Shuai Nie[2], Guangxu Ma[1,3], Johnny Vlaminck[4], Peter Geldhof[4], Nicholas A. Williamson[2], Gavin E. Reid[5,6,7], Robin B. Gasser[1]*

1 Department of Veterinary Biosciences, Melbourne Veterinary School, Faculty of Veterinary and Agricultural Sciences, The University of Melbourne, Parkville, Victoria, Australia, 2 Bio21 Mass Spectrometry and Proteomics Facility, The University of Melbourne, Parkville, Victoria, Australia, 3 College of Animal Sciences, Zhejiang Provincial Key Laboratory of Preventive Veterinary Medicine, Zhejiang University, Hangzhou, China, 4 Laboratory of Parasitology, Department of Virology, Parasitology and Immunology, Faculty of Veterinary Medicine, Ghent University, Merelbeke, Belgium, 5 School of Chemistry, The University of Melbourne, Parkville, Victoria, Australia, 6 Department of Biochemistry and Molecular Biology, The University of Melbourne, Parkville, Victoria, Australia, 7 Bio21 Molecular Science and Biotechnology Institute, The University of Melbourne, Parkville, Victoria, Australia

* robinbg@unimelb.edu.au

**Data Availability Statement:** All relevant data are within the manuscript and its Supporting Information files.

**Funding:** Research funding from the Australian Research Council (ARC) and National Health and

## Abstract

*Ascaris* is a soil-transmitted nematode that causes ascariasis, a neglected tropical disease affecting predominantly children and adolescents in the tropics and subtropics. Approximately 0.8 billion people are affected worldwide, equating to 0.86 million disability-adjusted life-years (DALYs). Exploring the molecular biology of *Ascaris* is important to gain a better understanding of the host-parasite interactions and disease processes, and supports the development of novel interventions. Although advances have been made in the genomics, transcriptomics and proteomics of *Ascaris*, its lipidome has received very limited attention. Lipidomics is an important sub-discipline of systems biology, focused on exploring lipids profiles in tissues and cells, and elucidating their biological and metabolic roles. Here, we characterised the lipidomes of key developmental stages and organ systems of *Ascaris* of porcine origin via high throughput LC-MS/MS. In total, > 500 lipid species belonging to 18 lipid classes within three lipid categories were identified and quantified–in precise molar amounts in relation to the dry weight of worm material–in different developmental stages/ sexes and organ systems. The results showed substantial differences in the composition and abundance of lipids with key roles in cellular processes and functions (e.g. energy storage regulation and membrane structure) among distinct stages and among organ systems, likely reflecting differing demands for lipids, depending on stage of growth and development as well as the need to adapt to constantly changing environments within and outside of the host animal. This work provides the first step toward understanding the biology of lipids in *Ascaris*, with possibilities to work toward designing new interventions against ascariasis.

## Author summary

Lipids are of vital importance in the biology of parasitic worms, particularly in relation to cellular membranes, energy storage, and intra- and intercellular signalling. However, very little is known about the biology of lipids in parasitic nematodes. Using a high-throughput LC-MS/MS approach, we characterised the first global lipidome for *Ascaris*. We

Medical Research Council (NHMRC) is gratefully acknowledged (RBG). The funders had no role in study design, data collection and analysis, decision to publish, or preparation of the manuscript.

**Competing interests:** The authors have declared that no competing interests exist.

investigated the lipid composition and abundance in key developmental stages/sexes as well as the organ systems of *Ascaris*. We observed substantial differences in lipid composition and abundance among these stages/sexes and among the organ systems studied. The findings provide a basis to start to understand lipid biology in *Ascaris*, with possible implications for developing new interventions against ascariasis.

## Introduction

Human ascariasis, caused by soil-transmitted nematode *Ascaris*, is one of the most important and commonest neglected tropical diseases [1,2]. Worldwide, ~ 800 million people are infected with this worm, with the highest prevalence in children and adolescents. This disease can cause chronic, long-term nutritional morbidity and affects cognitive development, equating to 0.86 million associated disability-adjusted life-years (DALYs) [3]. *Ascaris* of humans is very closely related to *Ascaris* of pigs [4], there is clear evidence of cross-transmission of these two operational taxonomic units (OTUs: pig-*Ascaris* and human-*Ascaris*) between humans and pigs and *vice versa* [5–8], and the species status of these OTUs is still controversial [9,10]. Given that pig-*Ascaris* is very closely related to the human-*Ascaris*, the *Ascaris*–swine model is ideal for detailed studies of *Ascaris* biology and ascariasis at the molecular level [11]. This model has enabled progress in the areas of genomics [11,12], transcriptomics [13,14] and proteomics [15,16], providing comprehensive information and resources to the scientific community and facilitating research into host-parasite interactions and immunobiology of *Ascaris*.

Lipids are a group of hydrophobic or amphipathic small molecules, which play key structural and biochemical roles in organisms. Increasing evidence indicates that pathogen-derived lipids, such as those of *Afipia felis*, *Mycobacterium tuberculosis* and *Schistosoma mansoni*, are actively involved in the biochemical interactions between pathogens and their hosts [17–20]. Therefore, gaining a better understanding of cellular lipid profiles of a parasite can assist in elucidating critical biological processes that occur during parasite invasion, establishment and persistence in host tissues, and in the discovery of diagnostic, drug and/or vaccine targets in parasitic worms, such as *Ascaris* [21]. Mass spectrometry-based lipidomic technology is enabling the detailed characterisation of lipid composition and abundance within organisms of interest under particular physiological or pathological conditions, and opens up an important avenue to undertake systems biological investigations [22].

To date, most lipidomic research of parasitic worms has focused on the flatworm *S. mansoni*. Using lipidomic tools, the biosynthesis and transport/incorporation of exogenous lipids in eggs and the adult female of *S. mansoni* have been studied [23,24]. Recently, advanced atmospheric pressure scanning microprobe matrix-assisted laser desorption/ionisation (AP-S-MALDI) mass spectrometry imaging (MSI) was used, for the first time, to characterise the lipid composition of *S. mansoni* tissues in a spatially-resolved manner [25]. However, a paucity of lipidomic work has been conducted on parasitic nematodes, with most studies focused predominantly on specific lipid categories or a single life-cycle stage [26,27]. Recently, we characterised the lipidome of *Haemonchus contortus* (barber's pole worm of ruminants) for six key developmental stages/sexes [28] and subsequently explored the lipidome of specific organs of the adult female of this species [29]. Using high throughput liquid chromatography-tandem mass spectrometry (LC-MS/MS), we were able, for the first time, to compare the relative quantitative differences of lipid species among different developmental stages/sexes or organ systems of a parasitic nematode. In the present study, we undertake the first semi-quantitative identification of lipids in the key developmental stages/sexes and in particular organ systems of adult pig-*Ascaris* (*A. suum*) using LC-MS/MS.

## Methods

### Ethics statement

*Ascaris suum* was produced in and collected from pigs using well-established methods [16] with animal ethics approval (permit no. EC2018/76) from Ghent University, Belgium.

### Procurement of *A. suum* stages, sexes and organ systems for lipidomic analysis

Samples of different developmental stages/sexes [i.e. infective third-stage (L3-egg), lung stage L3 (L3-lung) and fourth-stage (L4) larvae; female (Af) and male adults (Am)] and organ systems [i.e. reproductive and alimentary tracts, and body wall] were prepared from fresh, live *A. suum* using well-established methods [11,16] and washed extensively in physiological saline (5-times in 10 ml) to remove host-derived components; each alimentary tract was sliced longitudinally prior to washing. After the final wash, saline was removed with a pipette (200 μl tip) and individual samples were immediately snap frozen at −80˚C.

Developmental stages/sexes: L3-egg were obtained from fertile eggs isolated from the uteri of adult female worms and then cultured in 0.1% $K_2Cr_2O_7$ for 28–30 days at 25˚C; when 90% of the eggs were fully embryonated, L3-egg ($n \approx 5000$) were hatched from eggs [30] and then separated from egg-shell fragments and other debris by baermannisation [31]. L3-lung ($n \approx 3000$) were isolated/purified by baermannisation from the lungs of pigs orally inoculated with 500,000 infective eggs and euthanised after 7 days. L4s ($n \approx 800$) were isolated from the small intestines of pigs orally inoculated with 30,000 infective eggs and euthanised after 14 days. Af and Am worms were collected directly from the duodenum of infected pigs. In addition, organ systems were isolated from adult worms ($n = 3$ to 4 for each system) using a dissecting microscope: reproductive (FRT) and alimentary (FAT) tracts and body wall (FBW) from females; the reproductive (MRT) and alimentary (MAT) tracts and body wall (MBW) from males. Four replicate samples were collected for each developmental stage, sex and organ system.

Although it is not possible to be certain that all host-derived lipids were removed from *Ascaris* samples used for lipid extraction, every effort was made to eliminate host 'contamination', such that we are confident that minor host lipid contamination would not subsequently confound the lipidomic results of the present study. It would also not be possible to exclude the possibility that some host lipids had been absorbed via the intestinal tract of *Ascaris* and/or transported into tissues/cells of the parasite while in the host animal.

### Tissue homogenisation and lipid extraction

Samples ($n = 44$) were lyophilised in a benchtop, manifold freeze-drier prior to extraction, and then individually homogenised. Briefly, freeze-dried samples (4 mg each) were individually transferred to an Eppendorf tube (1.5 ml) containing 200 μl of ice-cold 40% (v/v) methanol, with 10 μl of isotope labelled internal lipid standards solution (330710X, Mouse SPLASH LIPIDOMIX, Merck, USA). Samples were homogenised with 100 μl of 0.5 mm zirconium oxide beads (ZROB05, Next Advance, USA) in a blender (Bead Bullet, Next Advance, USA) twice for 3 min. Blank tubes with water (included as controls) were processed in the same manner. Homogenised samples were processed using an established monophasic lipid extraction protocol [32] using 0.01% (v/v) butylated hydroxytoluene (= BHT) as an antioxidant to prevent lipid oxidisation throughout the extraction procedure. Briefly, 700 μl of 0.74/1/2 (v/v/v) water/chloroform/methanol plus BHT were added to each tube. The tubes were vortexed for 60 sec and incubated on a ThermoMix (Eppendorf, Germany) at 1,400 rpm for 5 min, and then centrifuged at 14,000 rpm and 22˚C for 15 min. The supernatant was then transferred to another

tube, and 400 μl of 1/2 (v/v) chloroform/methanol plus BHT were added, vortexed (60 s) and centrifuged as before. The supernatant was collected and pooled with the supernatant from the first extraction. The pooled lipid samples were then dried in a SpeedyVac, and each lipid pellet was re-suspended in 10 μl of 4/2/1 (v/v/v) isopropanol/methanol/chloroform plus BHT, and then diluted with 190 μl 100% methanol prior to analysis.

## High-performance liquid chromatography (HPLC) and mass spectrometric (MS) analyses

Samples (four replicates for each) were analysed by ultrahigh performance liquid chromatography (UHPLC) coupled to tandem mass spectrometry (MS/MS) employing a Vanquish UHPLC linked to an Orbitrap Fusion Lumos mass spectrometer (Thermo Fisher Scientific, San Jose, CA, USA), with separate runs in positive and negative ion polarities. Solvent A was 6/4 (v/v) acetonitrile/water and solvent B was 9/1 (v/v) isopropanol/acetonitrile; both solvents A and B contained 10 mM ammonium acetate. Each sample (10 μl) was injected into an RRHD Eclipse Plus C18 column (2.1 × 1000 mm, 1.8 μm; Agilent Technologies, USA) at 60˚C at a flow rate of 350 μl/min for 3 min using 30% solvent B. During separation, the percentage of solvent B was increased from 30% to 70% in 5 min, from 70% to 93% in 9 min, from 93% to 99% in 7 min, and from 91% to 97% in 31 min. Subsequently, the percentage of solvent B was increased to 99.5% in 0.1 min and then maintained at 99.5% for 4.9 min. Finally, the percentage of solvent B was decreased to 30% in 0.1 min and maintained for 3.9 min.

All MS experiments were performed using a Heated Electrospray Ionization (HESI) source. The spray voltages were 3.5 kV in positive ionisation-mode and 3.0 kV in negative ionisation-mode. In both polarities, the flow rates of sheath, auxiliary and sweep gases were 20 and 6 and 1 'arbitrary' unit(s), respectively. The ion transfer tube and vaporizer temperatures were maintained at 350˚C and 400˚C, respectively, and the S-Lens RF level was set at 50%. In the positive ionisation-mode from 3 to 24 min, top speed data-dependent scan with a cycle time of 0.6 s was used. Within each cycle, a full-scan MS-spectra were acquired firstly in the Orbitrap at a mass resolving power of 120,000 (at m/z 200) across an m/z range of 300–2000 using quadrupole isolation, an automatic gain control (AGC) target of 4e5 and a maximum injection time of 50 milliseconds, followed by higher-energy collisional dissociation (HCD)-MS/MS–a fragmentation technique which achieves a similar fragmentation pattern to that obtained using quadrupole-type collision induced dissociation–at a mass resolving power of 15,000 (at m/z 200), a normalised collision energy (NCE) of 27%, at positive mode and 30% at negative mode, an m/z isolation window of 1, a maximum injection time of 35 milliseconds and an AGC target of 5e4. For the improved structural characterisation of glycerophosphocholine (PC) lipid cations, a data-dependent product ion (m/z 184.0733)-triggered collision-induced dissociation (CID)-MS/MS scan was performed in the cycle using a q-value of 0.25 and a NCE of 30%, with other settings being the same as that for HCD-MS/MS. For the improved structural characterisation of TG lipid cations, the fatty acid + NH$_3$ neutral loss product ions observed by HCD-MS/MS were used to trigger the acquisition of the top-3 data-dependent CID-MS[3] scans (i.e. the 3 most abundant product ions of TG lipids identified by HCD-MS/MS were fragmented further by CID-MS[3]) in the cycle using a q-value of 0.25 and a NCE of 30%, with other settings being the same as that for HCD-MS/MS.

## Identification and quantification of lipids and statistical analysis

MS data were processed using LipidSearch software v.4.2.23 (Thermo Fisher Scientific, San Jose, CA, USA) [33]. Lipid ions were identified based on m/z values of precursor and fragmentation product ions. Key processing parameters were: target database, general; precursor

tolerance, 5 ppm; product tolerance, 10 ppm; product ion threshold, 1%; m-score threshold, 2; quantification m/z tolerance, ± 5 ppm; quantification retention time range, ± 1 min; use of main isomer filter and ID quality filters A, B and C; adduct ions, +H, +NH$_4$, +Na and +H-H$_2$O for positive ionisation mode, and −H, +CH$_3$COO, -2H and -CH$_3$ for negative ionisation mode. All lipid classes available were selected for the search (S1 Table). The same lipid annotations (within ± 0.1 min of retention time and 5 ppm of mass error) were merged into the aligned results. Unassigned peak areas were calculated for relative quantification and alignment. The shorthand notation used for lipid classification and structural representation follows the nomenclature proposed previously [34,35]. To filter false identifications, aligned lipids were examined manually. Lipids identified using LipidSearch were graded as follows: A = lipids with acyl/alkyl chains and class-specific fragmentation product ions; B = lipids with some acyl/alkyl chain-specific fragmentation product ions; C = lipids with acyl/alkyl chain-specific fragmentation product ions; and D = others.

For hexosyl ceramide (HexCer), monoacylglycerol (MG) and sphingomyelin (SM) lipids, a grade A-, B- or C-identification was required for at least two replicates of each sample, whereas for all other lipid classes, a grade A- or B-identification was required for at least two replicates of each sample. For all lipids, a ratio of MS/MS spectrum peaks assigned to the lipid among all fragmentation product ions > 40 was required in at least one sample group. All lipid LC-MS features were manually inspected and re-integrated when needed. The identification of lipid species was filtered further using lipid class-specific thresholds for apex intensity of extracted ion chromatogram (EIX) peak of lipid ion (S2 Table). Semi-quantitative analysis of lipid species was achieved by comparison of the identified peak areas against those of the correspondent isotope labelled internal lipid standards in the same lipid class, including diradylglycerols (DG), triradylglycerols (TG), glycerophosphatidic acid (PA), glycerophosphocholines (PC), glycerophosphoethanolamines (PE), glycerophosphoglycerols (PG), glycerophosphoinositols (PI), glycerophosphoserines (PS), lysoPC (LPC), lysoPE (LPE) and SM lipids, and reported as the amount of lipid per mg of dry weight. For the lipid classes without correspondent isotope-labelled internal lipid standards, the relative abundance of individual molecular species within these classes were normalised as follow: the MG species against the DG (18:1D7_15:0) internal standard; the cardiolipins (CL) and lysoPG (LPG) against the PG (18:1D7_15:0) internal standard; the lysoPI (LPI) against the PI (18:1D7_15:0) internal standard; the lysoPS (LPS) against the PS (18:1D7_15:0) internal standard; the ceramide (Cer) and HexCer against the SM (d36:2D9) internal standard. Given that only a single lipid standard per class was used, that identified lipids were normalised against a standard from a different class or sub-class, and that no attempts were made to quantitatively correct for different ESI responses of individual lipids due to concentration, acyl chain length, degree of unsaturation, or matrix effects caused by differences in chromatographic retention times compared with the relevant standards, the results reported here are semi-quantitative and should not be considered to reflect the absolute concentrations of each lipid or lipid sub-class [36]. UpSet plot of lipidomes were produced using the Intervene [37]. Principal component analysis (PCA) was conducted using Perseus software (v.1.6.1.1) [38,39]. One-way ANOVA Post Hoc Test was performed for multiple group comparisons by using GraphPad Prism 8.4.2 software (GraphPad, La Jolla, USA). Error bars represent the standard deviation (SD). Statistical significance was set at $P < 0.05$.

## Results

### Lipidome of developmental stages and organ systems of *Ascaris suum*

Using the high throughput LC-MS/MS approach, a total of 587 unique lipid species representing 18 lipid classes within three lipid categories, including glycerolipids (GL),

glycerophospholipids (GP) and sphingolipids (SP), were identified in all developmental stages and organ systems (*n* = 44 samples) of *A. suum* via LipidSearch (Table 1 and S3 Table). For all of these samples, the most commonly identified lipids were in the GL and GP categories, and represented 26% (*n* = 150) and 65% (*n* = 383) of lipid species. At the lipid-class level, triglyceride (TG) was the major class in *Ascaris* in the GL category, with a total of 143 identified, followed by the classes PC (*n* = 100), PE (*n* = 63) and PI (*n* = 63).

The numbers of lipids identified in the *A. suum* lipidome were highest in the L3 stage (L3-egg, *n* = 368; L3-lung, *n* = 336), followed by adult male (Am; *n* = 274), adult female (Af; *n* = 267) and the L4 stage (*n* = 237). The numbers of lipid species identified that were shared between or among developmental stages (i.e. L3-egg, L3-lung, L4, Af and Am) of *A. suum* are displayed in an UpSet plot (Fig 1). Most lipid species (*n* = 386; 66%) were detected in at least two developmental stages, and 126 (21%) were present in all developmental stages studied (Fig 1). By contrast, only small numbers of lipid species were uniquely detected in a particular stage. The L3-egg stage contained most of these lipids (*n* = 54; 9%), followed by L3-lung (*n* = 38, 6%) and Am (*n* = 19; 3%), whereas L4 and Af only had 4 unique lipids. All lipid species identified in each life stage are listed in S3 Table.

Similar to the situation for the developmental lipidome, most lipids were identified in at least two organ systems (i.e. reproductive tract, alimentary tract and body wall) of the adult worm (Fig 2). While > 22% of these lipids (*n* = 127) were detected in each organ system of adult male and female worm, < 15% were uniquely identified in each system. The greatest number of unique lipids identified was in MRT (*n* = 69), followed by FRT (*n* = 21), whereas no lipid was exclusively detected in MBW. All lipid species identified in individual organ systems from adult worms are listed in S3 Table.

**Table 1. Summary of the numbers of identified lipid species from lipid extracts derived from different developmental stages or organ systems of *Ascaris suum*.**

| Lipid category/class | Developmental stage | | | | | Organ system | | | | | | Total numbers |
|---|---|---|---|---|---|---|---|---|---|---|---|---|
| | L3-egg | L3-lung | L4 | Af | Am | FRT | MRT | FAT | MAT | FBW | MBW | |
| Glycerolipids | | | | | | | | | | | | |
| MG | 2 | 2 | 2 | 2 | 2 | 2 | 2 | 0 | 0 | 2 | 2 | 2 |
| DG | 1 | 0 | 1 | 5 | 4 | 4 | 4 | 5 | 3 | 3 | 4 | 6 |
| TG | 106 | 58 | 37 | 93 | 75 | 104 | 56 | 86 | 56 | 93 | 81 | 142 |
| Glycerophospholipids | | | | | | | | | | | | |
| PA | 7 | 8 | 4 | 10 | 9 | 8 | 9 | 14 | 13 | 12 | 13 | 17 |
| PC | 63 | 47 | 31 | 37 | 37 | 62 | 73 | 65 | 74 | 49 | 32 | 100 |
| PE | 35 | 36 | 19 | 25 | 32 | 16 | 34 | 26 | 26 | 27 | 33 | 63 |
| PG | 6 | 6 | 5 | 5 | 5 | 2 | 5 | 4 | 4 | 5 | 5 | 6 |
| PI | 40 | 52 | 43 | 32 | 30 | 38 | 47 | 45 | 47 | 29 | 22 | 63 |
| PS | 23 | 17 | 6 | 7 | 11 | 5 | 24 | 6 | 8 | 11 | 7 | 35 |
| CL | 39 | 38 | 22 | 19 | 20 | 25 | 34 | 20 | 18 | 30 | 20 | 45 |
| LPC | 11 | 24 | 20 | 10 | 12 | 12 | 24 | 12 | 13 | 11 | 9 | 24 |
| LPE | 8 | 10 | 7 | 7 | 6 | 9 | 8 | 7 | 6 | 4 | 4 | 14 |
| LPG | 0 | 1 | 1 | 0 | 0 | 0 | 0 | 0 | 0 | 0 | 0 | 1 |
| LPI | 5 | 8 | 9 | 1 | 1 | 5 | 5 | 4 | 6 | 2 | 1 | 9 |
| LPS | 4 | 6 | 3 | 2 | 3 | 2 | 5 | 2 | 2 | 1 | 0 | 6 |
| Sphingolipids | | | | | | | | | | | | |
| SM | 9 | 4 | 8 | 3 | 9 | 5 | 8 | 2 | 3 | 4 | 8 | 12 |
| Cer | 7 | 16 | 15 | 7 | 14 | 6 | 13 | 10 | 17 | 9 | 12 | 30 |
| HexCer | 2 | 3 | 4 | 2 | 4 | 2 | 5 | 5 | 7 | 2 | 2 | 12 |
| Totals | 368 | 336 | 237 | 267 | 274 | 307 | 356 | 313 | 303 | 294 | 255 | 587 |

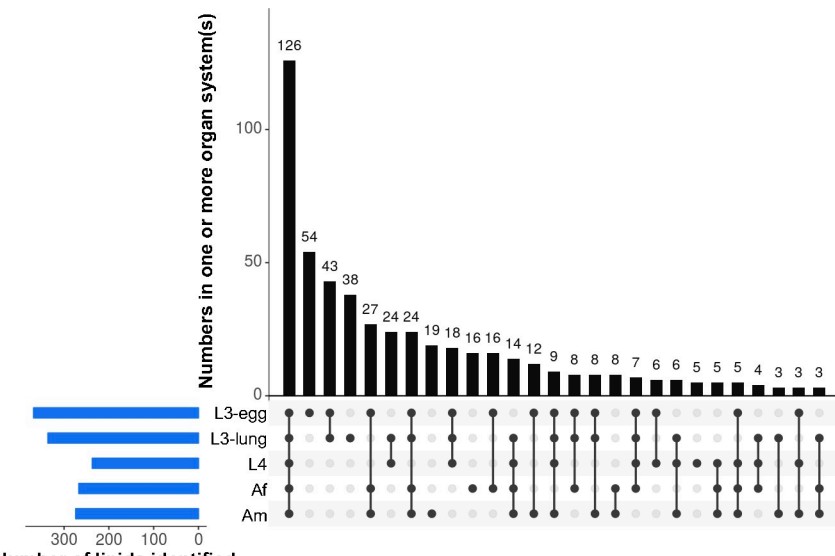

**Fig 1. The presence and numbers of lipids identified in five different developmental stages/sexes of *Ascaris suum* (i.e. infective third-stage (L3-egg), lung stage L3 (L3-lung) and fourth-stage (L4) larvae; female (Af) and male adults (Am)).** Connected dots display shared lipid species between or among developmental stages/sexes, and the total number of lipid species in a particular stage/sex is shown (bottom left corner).

## Fatty acyl composition of the lipidome of *Ascaris suum*

The analysis of fatty acyl compositions revealed that the lipidome of *A. suum* contained a large section of long-chain (>12 carbons), even-numbered and unsaturated fatty acyls, which contributed 97%, 88% and 62% to the overall fatty acyl composition, respectively (Table 2). In

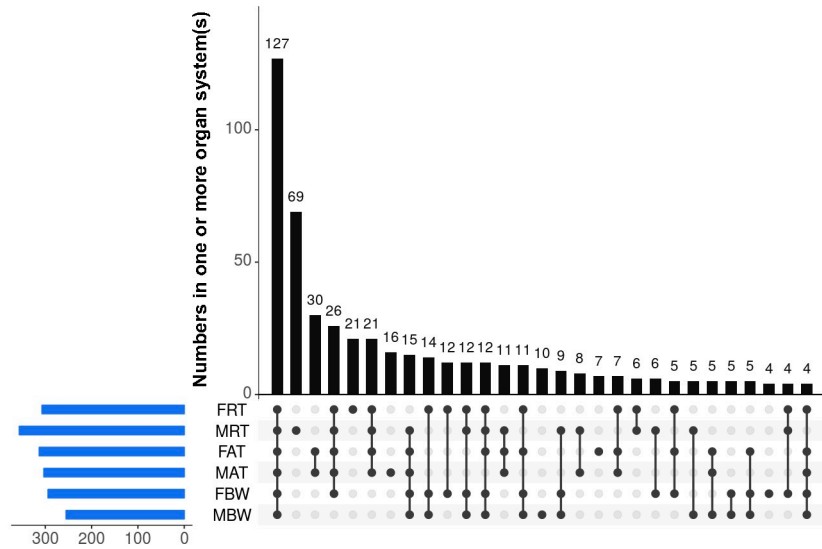

**Fig 2. The presence and numbers of lipids identified in six distinct organ systems of *Ascaris suum* adults (i.e. the reproductive (FRT) and alimentary (FAT) tracts and body wall (FBW) of female adults; the reproductive (MRT) and alimentary (MAT) tracts and body wall (MBW) of male adults).** Connected dots display lipid species shared between or among the different organ systems, and the total numbers of lipid species per organ system is shown (bottom left).

**Table 2. Fatty acyl (FA) composition of identified lipid species in the lipidome of *Ascaris suum*.**

| Lipid category | Saturated (%) | | Unsaturated (%) | | Odd-numbered chain FA (%) | Even-numbered chain FA (%) | Total no. of FAs |
|---|---|---|---|---|---|---|---|
| | Medium-chain FA [1] | Long-chain FA [2] | Medium-chain FA | Long-chain FA | | | |
| Glycerolipids | 8.2 | 33.9 | ND | 58.0 | 10.6 | 88.7 | 440 |
| Glycerophospholipids | 0.4 | 32.7 | ND | 66.9 | 9.2 | 90.8 | 804 |
| Sphingolipids | ND | 59.4 | ND | 40.6 | 36.5 | 63.5 | 96 |
| Totals | 2.9 | 35.0 | ND | 62.1 | 11.6 | 88.4 | 1340 |

[1] Medium-chain FA contains 6–12 carbons; [2] Long-chain FA contains > 12 carbons; [3] ND, not detected

total, 64 saturated lipid species of 10 classes were observed in the lipidome. Most of them belonged to Cer ($n = 12$), PC ($n = 10$), TG ($n = 10$), PE ($n = 8$), LPE ($n = 8$) and LPC ($n = 8$). Additionally, 93 ether-linked lipids were detected; they were predominantly found in the GP category ($n = 79$), including classes PC ($n = 30$), PE ($n = 26$) and PI ($n = 9$), whereas the remainder of ether-linked lipids ($n = 14$) were within the GP category (i.e. TG class).

## Alterations in lipid abundance during development

PCA of the developmental lipidome of *A. suum* showed that the difference in the quantity of lipids among five developmental stages/sexes (i.e. L3-egg, L3-lung, L4, Af and Am) was greater than variation within a particular stage (i.e. among four replicates) (Fig 3). The two-dimensional diagram (Fig 3) reveals a clear division of the lipidomic data set into three distinct groups, corresponding to L3-egg, L3-lung and the intestinal stages (i.e. L4, Af and Am). Interestingly, limited variation in lipid amount was observed among adult stages. Of all five developmental stages/sexes, the largest amount of total lipids was measured in third-stage larvae (i.e. L3-egg and L3-lung) (Fig 4A). The semi-quantitative analysis revealed that third-stage larvae (i.e. L3-egg and L3-lung) contained > 150 and 100 μM/mg (micromole of lipids per milligram of dry worm body weight), respectively, whereas ≤ 35 μM/mg were measured in other developmental stages/sexes (i.e. L4, Af and Am) (Fig 4A).

As expected, lipid categories GP ($n = 155$ to 253) and GL ($n = 44$ to 109), for which a large number of lipids species were identified (Table 1), contributed predominantly to the lipid

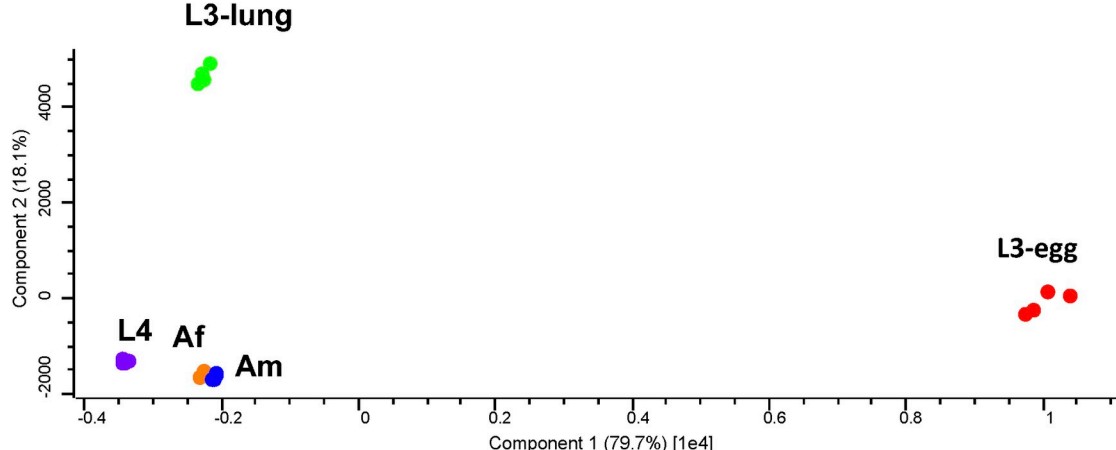

**Fig 3. Principal component analysis of the developmental lipidome of *Ascaris suum* representing infective third-stage (L3-egg), lung stage L3 (L3-lung) and fourth-stage (L4) larvae; female (Af) and male adults (Am).**

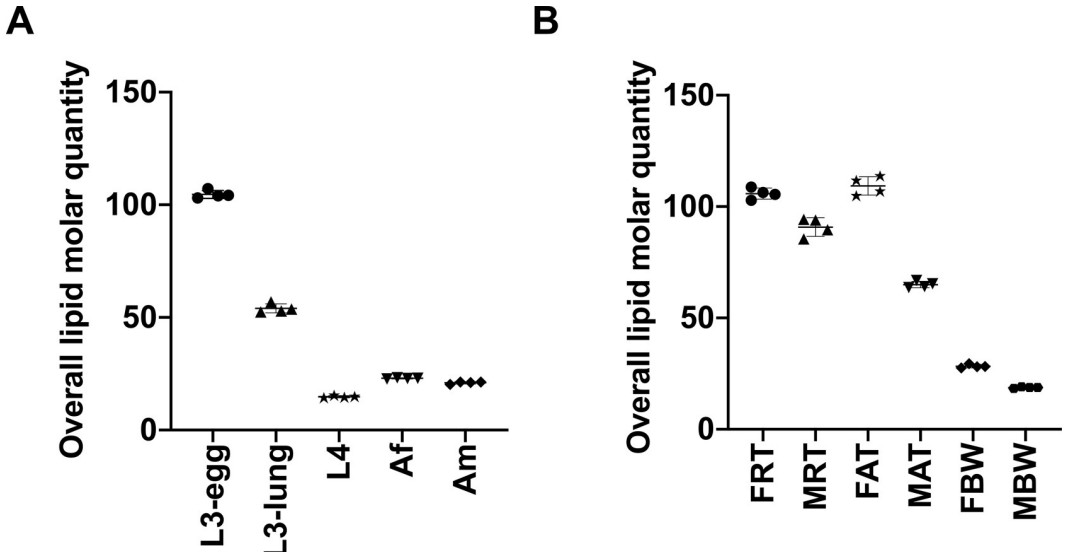

**Fig 4. Quantitative differences in total lipids among developmental stages/sexes (A) and among adult organ systems (B) of *Ascaris suum*.** The five developmental stages/sexes studied were: infective third-stage (L3-egg), lung stage L3 (L3-lung) and fourth-stage (L4) larvae; female (Af) and male adults (Am). The six organ systems studied were: the reproductive (FRT) and alimentary (FAT) tracts and body wall (FBW) of female adults; the reproductive (MRT) and alimentary (MAT) tracts and body wall (MBW) of male adults. Semi-quantitative analysis of lipid species, reported as the amount of lipid per mg of dry weight (μM/mg). Statistical analysis was performed by ANOVA (* $P < 0.05$). Error bars indicate ± SD (four replicates).

abundance in *A. suum* across five key developmental stages/sexes (Fig 5 and S1 Fig). The overall GP abundance reached a peak in third-stage larvae (i.e. L3-egg and L3-lung), was significantly lower in later developmental stages/sexes (i.e. L4, Af and Am). Membrane structure-related lipid classes, such as PC, PE and PI, contributed significantly to a low overall GP abundance (Fig 6 and S2 Fig). Individual PC, PE and PI lipid species, such as PC (36:3), PE (O-36:1) and PI (38:4), which contained even-numbered fatty acyl chains (e.g., 18:0, 18:1 or 18:2) predominated in the third-stage larvae (S3 Table). Additionally, LPC, LPE, LPG and LPS classes peaked in L3-lung, and then were substantially reduced during the migration from lung (i.e. L3-lung) to the small intestine (i.e. L4, Af and Am) (S2 Fig). Within the GL category, a significantly higher abundance of TG was measured in L3-egg compared with all other stages/sexes studied (Fig 6). Notably, TG exhibited a slightly higher level in Af than in Am. Deeper analysis of individual lipid species showed that TG lipids with C18 fatty acyl chains (e.g. 18:0, 18:1, 18:2 and 18:3) predominated and that many of these lipids (*n* = 9) showed a high abundance (> 1 μM/mg) for the TG class (S3 Table). Nevertheless, a significantly higher level of total saturated lipid was observed in L3-lung, whereas high levels of ether-linked lipid were detected in the L3-egg and L3-lung (Fig 7). All lipid classes and individual lipid species as well as their differences in abundance among developmental stages/sexes are given in S2–S4 Figs and S3 Table.

## Difference in lipid abundance among organ systems of adult *A. suum*

The two-dimensional PCA showed that lipidomic data of organ systems for male adult *Ascaris* (i.e. MRT, MAT and MBW) and the body wall from the female worm (i.e. FBW) clustered tightly together, to the exclusion of the reproductive and alimentary tracts of female *Ascaris* (i.e. FRT and FAT) (Fig 8). Semi-quantitative analysis of the organ systems showed an enrichment of total lipids in the reproductive and alimentary tracts of adult *Ascaris* (Fig 4B). FRT

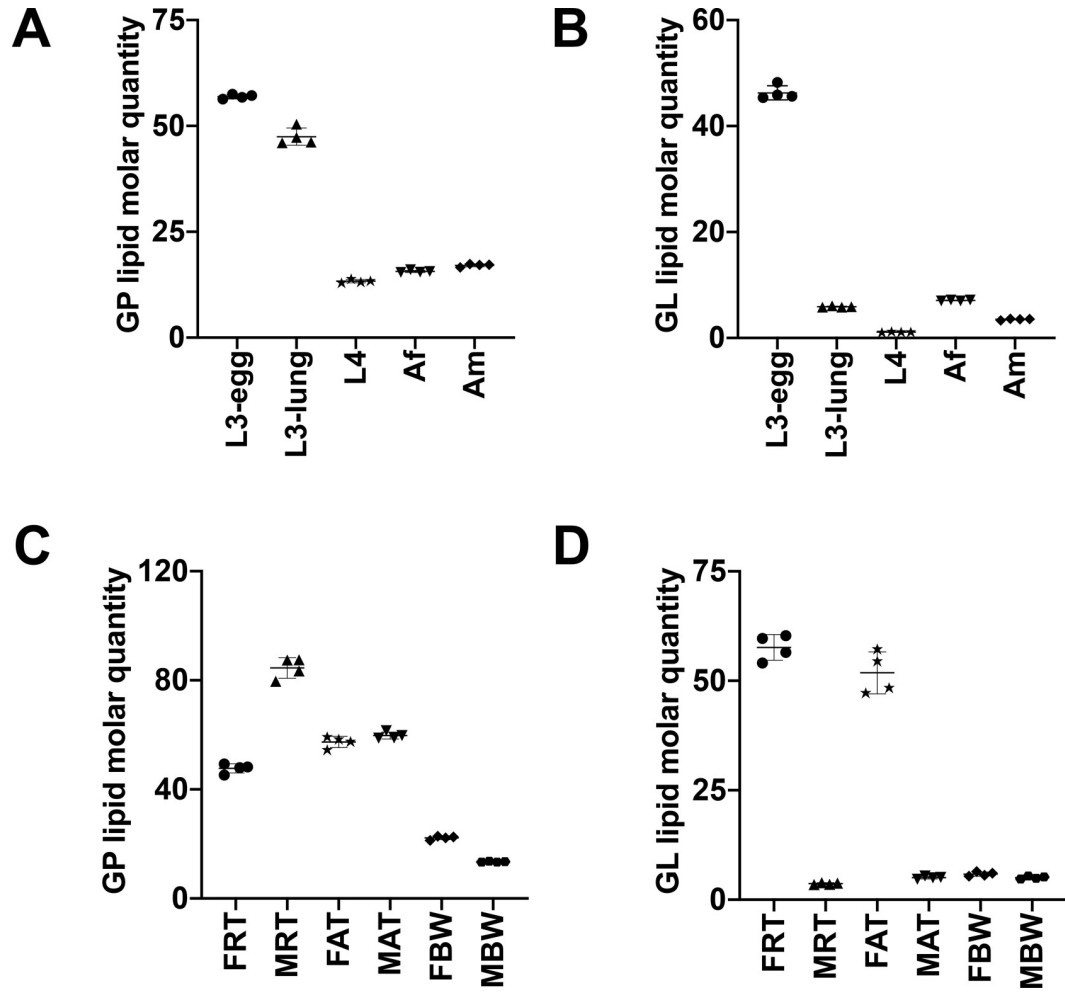

**Fig 5. Quantitative differences in glycerophospholipids (GP) and glycerolipids (GL) categories among developmental stages/sexes (A and B) and among adult organ systems (C and D) of *Ascaris suum*.** The five developmental stages/sexes studied were: infective third-stage (L3-egg), lung stage L3 (L3-lung) and fourth-stage (L4) larvae; female (Af) and male adults (Am). The six adult organ systems studied were: the reproductive (FRT) and alimentary (FAT) tracts and body wall (FBW) of female adults; the reproductive (MRT) and alimentary (MAT) tracts and body wall (MBW) of male adults. Semi-quantitative analysis of lipid species, reported as the amount of lipid per mg of dry weight (μM/mg). Statistical analysis was performed by ANOVA (* $P < 0.05$). Error bars indicate ± SD (four replicates).

(141 μM/mg) had > 4 times more lipid overall as compared with MBW (32 μM/mg). Except for the reproductive tract, a comparisons of the same organ system between the sexes showed that the female worm had more total lipids than the male. Similar to the developmental lipidome, GP (ranged 27–134 μM/mg) and GL (ranged 3–49 μM/mg) were the two most abundant lipid categories in *A. suum* at an organ system level, whereas only small amounts (range: 1–11 μM/mg) of lipids of the SP category were detected (Fig 5 and S1 Fig).

Regarding the reproductive tract, the overall GP abundance was significantly higher in male (134 μM/mg) than in female (69 μM/mg) (Fig 5B). In contrast, GL abundance showed the opposite trend, with significantly higher levels in FRT (49 μM/mg) than in MRT (3 μM/mg) (Fig 5D). A deeper analysis of the lipidomic data set according to organ system revealed differences primarily in the lipids in the PC, PE and TG classes, characterised by a higher abundance of individual PC species (e.g. PC (16:0_20:4), PC (18:0e_20:2), PC (18:1_20:2) and PC (20:1_18:2)) and PE species (e.g. PE (O-18:0_18:1), PE (O-18:0_20:1)) in FRT compared

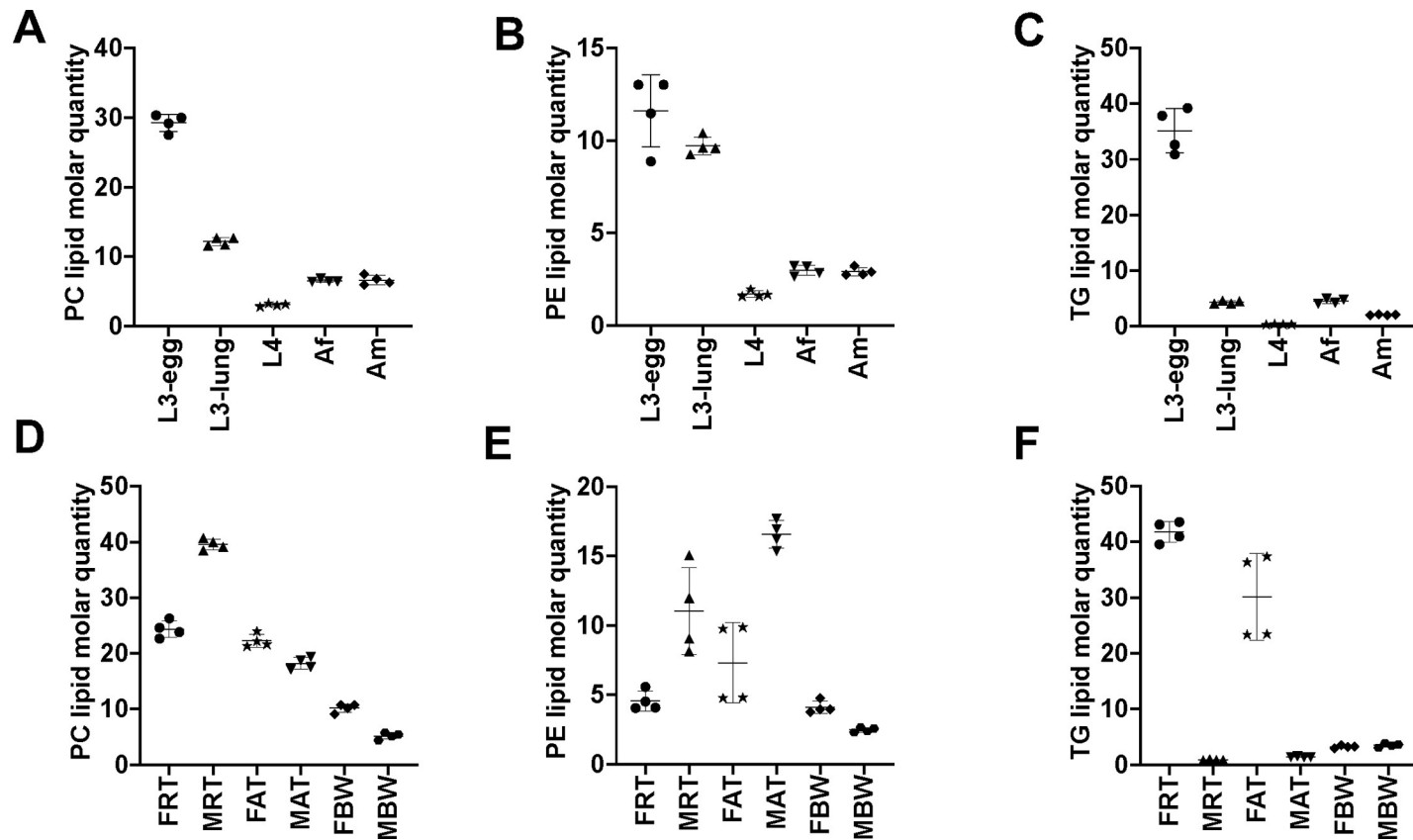

**Fig 6. Quantitative differences in PC, PE and TG among developmental stages/sexes (A, B and C) and among adult organ systems (D, E and F) of *Ascaris suum*.** The five developmental stages/sexes studied were: infective third-stage (L3-egg), lung stage L3 (L3-lung) and fourth-stage (L4) larvae; female (Af) and male adults (Am). The six adult organ systems studied were: the reproductive (FRT) and alimentary (FAT) tracts and body wall (FBW) of female adults; the reproductive (MRT) and alimentary (MAT) tracts and body wall (MBW) of male adults. Semi-quantitative analysis of lipid species, reported as the amount of lipid per mg of dry weight (μM/mg). Statistical analysis was performed by ANOVA (* $P < 0.05$). Error bars indicate ± SD (four replicates).

with MRT; and a higher abundance of TG species, such as TG (17:0_18:2_6:0), TG (18:0_18:1_18:2) and TG (18:1_18:1_6:0), in MRT (S3 Table).

In the alimentary tract, the total GL amount was more abundant in female (36 μM/mg) than in male (3 μM/mg), whereas the overall GP in the alimentary tract was at a comparable level (nearly 60 μM/mg) in female and male. Notably, individual TG species with even-numbered fatty acyl chains, such as TG (16:0_16:1_18:1), TG (16:1_18:1_18:2) and TG (18:0_18:1_18:2), differed markedly in abundance between FAT and MAT. There was no significant difference in the abundance of overall GP and GL in the body wall. Subsequent analyses revealed that both saturated and either-linked lipids were highly abundant in the reproductive and alimentary tracts of both female and male worms, with no significant difference between the sexes. All lipid classes and individual lipid species as well as their abundance levels in the organ systems are displayed in S2–S4 Figs and S3 Table.

## Discussion

The life cycle of *Ascaris* is direct but complex: in the initial stage of infection, the invading larvae undergo a hepato-pulmonary migration, passing through the intestinal tissues, liver and lungs, and eventually returning to and establishing in the intestinal tract of the host [4]. Such tissue migration requires *Ascaris* to adapt to constantly changing external environments, in

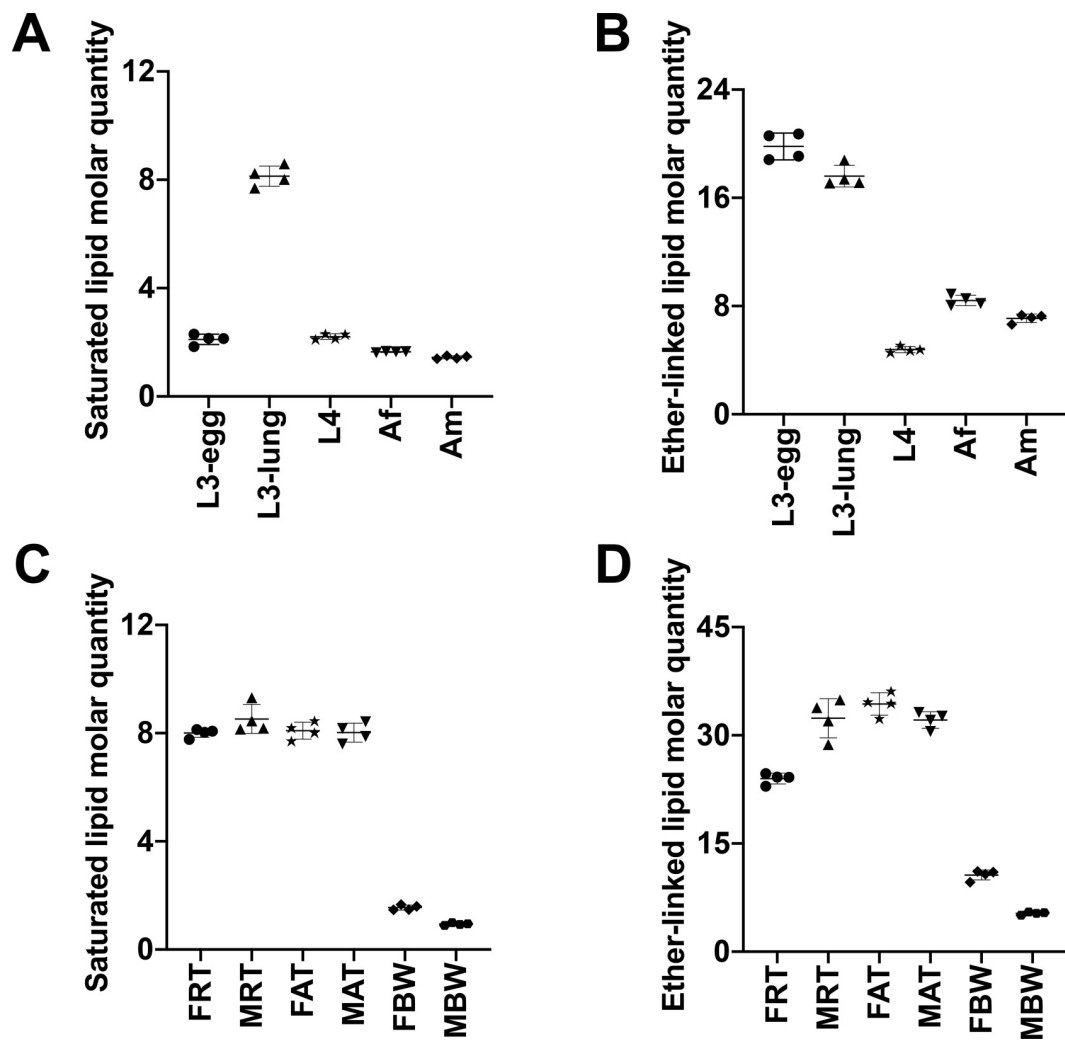

**Fig 7. Quantitative differences in total saturated fatty acyl- and ether-linked lipid among developmental stages/sexes (A and B) and among adult organ systems (C and D) of *Ascaris suum*.** The five developmental stages/sexes studied were: infective third-stage (L3-egg), lung stage L3 (L3-lung) and fourth-stage (L4) larvae; female (Af) and male adults (Am). The six adult organ systems studied were: the reproductive (FRT) and alimentary (FAT) tracts and body wall (FBW) of female adults; the reproductive (MRT) and alimentary (MAT) tracts and body wall (MBW) of male adults. Semi-quantitative analysis of lipid species, reported as the amount of lipid per mg of dry weight (μM/mg). Statistical analysis was performed by ANOVA (* $P < 0.05$). Error bars indicate ± SD (four replicates).

terms of temperature, oxygen supply, redox potential and host immune reactions [40,41]. In this lipidomic study, we observed substantial differences in the composition and abundance of lipids, likely associated with key biological functions, including energy metabolism and membrane structure during the parasite's growth and development. Alterations in lipidomic profiles are likely reflected in the adaptation of this nematode to changing environments within the host animal.

Lipid classes PC and PE had the highest representation in individual developmental stages/sexes and organ systems (i.e. the reproductive and alimentary tracts) studied. Their abundance in these samples is likely associated with them being common building blocks of membrane bilayers [42]. Similar to previous studies of parasitic nematodes [26–28], we showed that the amounts of PC were significantly higher than PE in the lipidome of *A. suum*. In addition, the

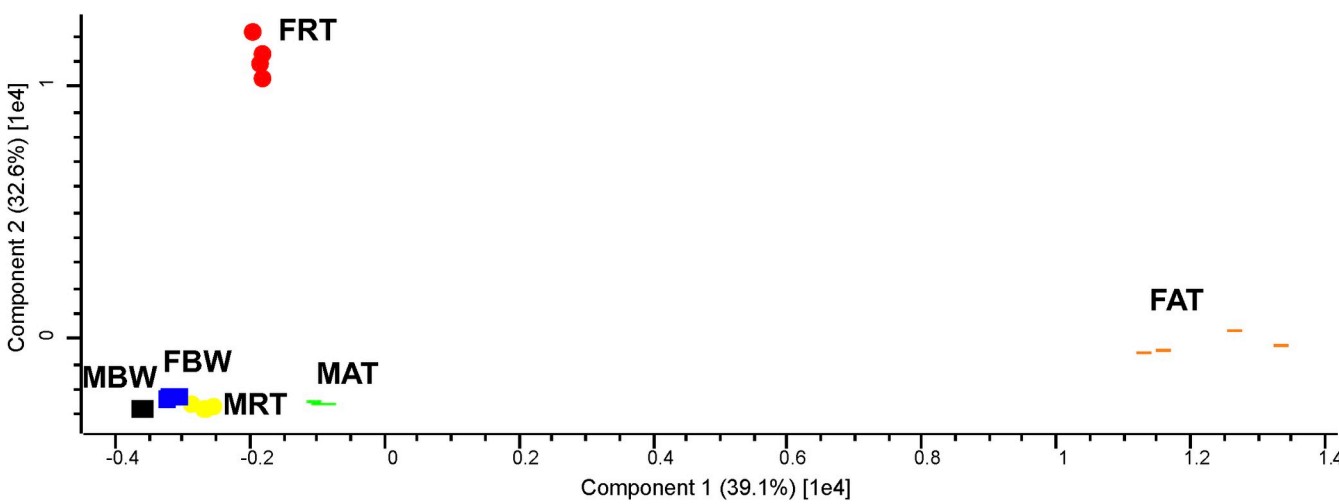

**Fig 8. Principal component analysis of the lipidomic data for *Ascaris suum* organ systems: the reproductive (FRT) and alimentary (FAT) tracts and body wall (FBW) of female adults; the reproductive (MRT) and alimentary (MAT) tracts and body wall (MBW) of male adults.**

reproductive tract of male *Ascaris* contained PC and PE in higher amounts than in that of the female, indicating a potentially higher portion of respective membranes in the male reproductive system. Notably, the body wall of *Ascaris* contained a low level of lipids, including PC. This observation contrasts findings from studies of *S. mansoni*, which revealed an enrichment of PC in the tegument (exterior surface of the worm) compared with parenchymal tissues [23], which is explained by the differing body compositions of pseudocoelomates (including nematodes), with an external cuticle, consisting of mainly collagens, cuticlins and glycoproteins [43], compared with acoelomate worms, which have an external tegumental layer and internal parenchyma [44].

Generally, it is believed that the stability and fluidity of membranes are influenced by the ratio of saturated to unsaturated lipids [42,45]. Lipids with double-bond unsaturated fatty acyl, which have lower melting points, allow associated membranes to maintain their fluidity in cold environmental temperatures. Previous studies showed that the free-living nematode, *Caenorhabditis elegans*, can increase the abundance of unsaturated membrane lipids to cope with cooler environments [46] and reduce them accordingly to adapt to increased temperatures [47]. In agreement with this observation, the present study revealed that saturated lipids of *Ascaris* belonged mainly to the GP lipid category, which includes PC ($n$ = 10), PE ($n$ = 8), LPC ($n$ = 8), LPE ($n$ = 8) and LPI ($n$ = 8) (S3 Table). In addition, the amount of these saturated lipids only represents 0.8% of the overall lipid abundance in the larval stage from the egg (i.e. L3-egg; usually at an environmental temperature of 5–25˚C), whereas this ratio was higher (range: 3.3–6.8%) in parasitic stages that live within the host animal (i.e. L3-lung, L4, Af and Am; 38–40˚C), suggesting an adaption of *Ascaris* to environmental temperature during development. However, contrary findings have been documented for other parasitic nematodes studied, including *H. contortus*, *Dictyocaulus viviparus* [48], *Oesophagostomum dentatum* and *Oesophagostomum quadrispinulatum* [49]. Becker et al. [48] proposed that the differing abundance of saturated lipids during development between free-living nematodes and parasitic nematodes might be due to variations in the adaptability of these distinct worms to stresses (e.g., temperature and desiccation) and/or to their mode of nutrient/molecular uptake via the cuticle (cf. [50,51]). However, much more work is needed to provide clear evidence to support this proposal. The recent use of advanced mass spectrometry imaging (i.e. high-resolution

atmospheric pressure scanning microprobe matrix-assisted laser desorption/ionisation mass spectrometry imaging) to characterise the lipidome of a single adult stage of *Schistosoma mansoni* signals an opportunity to elucidate the spatial distribution of membrane lipids in parasitic worms [25].

In addition to saturated lipids, the presence of ether-linked lipids also significantly affect the stability and fluidity of membrane [52]. Lipids with an alkyl chain attached to the sn-1 position by an ether bond–which provides stronger intermolecular hydrogen bonding between the headgroups–affect membranes fluidity and dynamics [52]. Previous studies of *S. mansoni* showed that this parasitic worm tends to enrich membrane ether-linked phospholipids to increase the stability and relatively inertia of membrane, in order to minimise degradation by host lipases [53,54]. Interestingly, we observed a higher ratio of ether-linked lipids in total lipids of *A. suum* stages within the host animal (i.e. L3-lung, L4, Af and Am; range: 13–25%) compared with the L3-egg stage (10%), suggesting a similar adaptation of *Ascaris* to the host environment.

Apart from the stability and fluidity of membranes, parasite-derived glycerophospholipids (i.e. GP) can be actively released from worms, such as *A. suum* and filarioids, into host tissues or tissue spaces [21,26], and likely play key roles in host-parasite interactions [55]. Evidence for *S. mansoni*, for example, showed that parasite-derived glycerophospholipids (e.g. LPC and LPS) can stimulate Toll-like receptor-2-dependent mechanisms to suppress the cytokine production and eosinophil activation [17,18]. To be able to effectively interact with a host, such parasite-derived glycerophospholipids need to be delivered by lipid-binding proteins. Interestingly, two mass spectrometry-based studies [16,56] revealed a series of lipid binding associated proteins (e.g. fatty acid and retinoid-binding proteins, vitellogenin and PE-binding proteins) in the soluble excretory-secretory products and membrane-enriched extracellular vesicles from parasitic stages of *A. suum*, indicating that such parasite-derived lipids might be involved in parasite-host cross-talk.

Interestingly, the findings of the present study are consistent with previous studies of other parasitic nematodes, including *Ancylostoma caninum*, *D. viviparus* and *H. contortus*, that TG abundance levels significantly decrease throughout the development of a nematode [28,48,57]. In egg and early larval stages, high ATP levels and adequate energy reserves are likely required to ensure proper growth and development, as these particular stages do not always have access to an abundance of nutrients from the environment and/or host. A previous study [58] showed high-level of activities of the enzymes in the glyoxylate cycle and the beta-oxidation pathway of the dauer larva of *C. elegans*, indicating the importance of lipid stores as a primary energy reserve in this stage. It has been proved that early egg and larval stages of *Ascaris* enable the conversion of endogenous maternally-accumulated lipids into succinate and malate for ATP production via the same metabolic pathway (i.e. beta-oxidation and glyoxylate cycle) [59]. The reduction of TG lipids during *Ascaris* development is most likely due to an exhaustion of lipid reserves. On the other hand, the reproductive tract of the gravid female adult of *Ascaris* contains TG in greater amounts than other organ systems, a finding that is consistent with the lipidomic study of the same organ system in gravid female *H. contortus* [29]. Given the anaerobic or microaerobic environment in the gut lumen and a lack of essential oxidation enzymes, adult stages of parasitic nematodes appear to be incapable of converting exogenous lipids into a direct energy supply via fatty acid beta-oxidation. It is evident that parasitic nematodes actively store large amounts of lipids in the egg stage, so that early larval stages get a "head start" and are not reliant on other sources in the initial phase of the life cycle. In the free-living nematode, *C. elegans*, a similar accumulation of TG occurs in embryos as a result of feeding on bacteria, with a peak around egg laying [60].

## Conclusions

In the present study, we used high throughput LC-MS/MS, employing isotope-labelled standards, to characterise the global lipidome of *A. suum*. More than 500 lipids were quantified in precise molar amounts in relation to respective dry weights of worm material. This is the first lipidomic study of a parasitic nematode using an accurate, semi-quantitative analytical approach–which normalises to dry weight rather than to total lipid intensity. The established workflow provides a cost-effective way to undertake comprehensive lipidomic analyses of parasites and provides an opportunity to quantitatively define lipidomic profiles in the best characterised nematode, *C. elegans*. This lipidomic data set extends our knowledge of lipid biology of parasitic worms and has the potential to assist the discovery of new interventions.

## Supporting information

**S1 Table. The full list of lipid classes selected for subsequent lipid identification/annotation using LipidSearch software v.4.2.23.**
(XLSX)

**S2 Table. The thresholds for apex intensity of extracted ion chromatogram (EIX) peak of lipid ion of each lipid class.**
(XLSX)

**S3 Table. Full list of identified lipid species in all developmental stages and organ systems (*n* = 44 samples) of *Ascaris suum*.**
(XLSX)

**S1 Fig.** Quantitative differences in sphingolipids (SP) among developmental stages/sexes (A) and among adult organ systems (B) of *Ascaris suum*.
(TIF)

**S2 Fig. Quantitative differences in PA, PG, PI, PS, CL, LPC, LPE, LPG, LPI and LPS among different life stages/sexes and among adult organ systems of *Ascaris suum*.**
(TIF)

**S3 Fig. Quantitative differences in MG and DG among different life stages/sexes and among adult organ systems of *Ascaris suum*.**
(TIF)

**S4 Fig. Quantitative differences in SM, Cer and HerCer among different life stages/sexes and among adult organ systems of *Ascaris suum*.**
(TIF)

## Acknowledgments

The authors thank Dr Vinzenz Hofferek (School of Chemistry, The University of Melbourne) for his kind support with lipid extraction.

## Author Contributions

**Conceptualization:** Tao Wang, Robin B. Gasser.

**Data curation:** Tao Wang, Shuai Nie, Guangxu Ma.

**Formal analysis:** Tao Wang, Shuai Nie.

**Funding acquisition:** Peter Geldhof, Gavin E. Reid, Robin B. Gasser.

**Investigation:** Tao Wang, Shuai Nie, Guangxu Ma.

**Methodology:** Tao Wang, Shuai Nie, Gavin E. Reid.

**Project administration:** Tao Wang.

**Resources:** Johnny Vlaminck, Peter Geldhof, Robin B. Gasser.

**Software:** Tao Wang, Shuai Nie.

**Supervision:** Robin B. Gasser.

**Validation:** Shuai Nie, Gavin E. Reid.

**Visualization:** Tao Wang.

**Writing – original draft:** Tao Wang, Robin B. Gasser.

**Writing – review & editing:** Tao Wang, Shuai Nie, Guangxu Ma, Johnny Vlaminck, Peter Geldhof, Nicholas A. Williamson, Gavin E. Reid, Robin B. Gasser.

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
