## [Decision Letter · Decision Letter 0]

21 Sep 2020

Dear Dr Wang,

Thank you very much for submitting your manuscript "Quantitative lipidomic analysis of Ascaris suum" for consideration at PLOS Neglected Tropical Diseases. As with all papers reviewed by the journal, your manuscript was reviewed by members of the editorial board and by several independent reviewers. The reviewers appreciated the attention to an important topic. Based on the reviews, we are likely to accept this manuscript for publication, providing that you modify the manuscript according to the review recommendations. 

Sincerely,

Michael Cappello

Associate Editor

Sara Lustigman

Deputy Editor

Reviewer's Responses to Questions

**Key Review Criteria Required for Acceptance?**

**Methods**

-Are the objectives of the study clearly articulated with a clear testable hypothesis stated?

-Is the study design appropriate to address the stated objectives?

-Is the population clearly described and appropriate for the hypothesis being tested?

-Is the sample size sufficient to ensure adequate power to address the hypothesis being tested?

-Were correct statistical analysis used to support conclusions?

-Are there concerns about ethical or regulatory requirements being met?

Reviewer #1: Clearly described and appropriate.

Reviewer #2: The methods are clear, logical and scientifically sound. However, some more description of how identifications were made would be helpful.

**Results**

-Does the analysis presented match the analysis plan?

-Are the results clearly and completely presented?

-Are the figures (Tables, Images) of sufficient quality for clarity?

Reviewer #1: Descriptive but comprehensive

Reviewer #2: The results are interpreted in a logical fashion.

**Conclusions**

-Are the conclusions supported by the data presented?

-Are the limitations of analysis clearly described?

-Do the authors discuss how these data can be helpful to advance our understanding of the topic under study?

-Is public health relevance addressed?

Reviewer #1: The lipidomic analysis of Ascaris suum by Wang et al provides a unique qualitative characterization of various stages and tissue compartments of larval and adult male and female worms that is important as a baseline resource. The work is largely descriptive with no hypothesis testing but is nevertheless important in comparison to other free living and parasite nematode and helminth worms. 

Specific criticisms:

The L4 and adult stages in the intestinal lumen have an acquired microbiome and, as luminal feeders, ingest host epithelial cells that are shed in large numbers into the intestine. The analysis does not attempt to distinguish host and microbiome lipids from that of the worm; which could be an important distinction in strategies to interfere with Ascaris development and residence in the host intestine.

The pseudo-coelomic or peri-enteric fluid of adult worms is a major source of lipid containing molecules that may differ between adult male and female worms, but it was not characterized in this study. This is curious since it is readily available and easy to isolate. Please provide an explanation for the absence of distinct data on this worm compartment.

Reviewer #2: The conclusions are justified by the data presented.

**Editorial and Data Presentation Modifications?**

Reviewer #1: minor revisions

Reviewer #2: I only have a few suggestions for the authors to consider.

1. It might be a good idea to define the authors meaning of "identified" when the term is first used.

2. In the Methods or Results section some description of how identifications were confirmed would be helpful for the reader to be able to assess the reliability of "confident identifications". I am left guessing as to how identifications were made other than by measured mass. What type of fragments were matched in MS/MS spectra and to what?

3. I am ignorant of the lipidome of nematodes, but do they make sterols? Perhaps some comment regarding other lipid classes which were not analysed is warranted, particularly as there is much discussion of membrane fluidity.

Minor points.

1. What was the logic behind adding BHT at the very end of the sample preparation protocol?

2. For those not using Thermo instruments it would be good learn the difference between CID and HCD.

3. Similarly, some explanation of top-3 data dependent MS3 scans would be useful.

**Summary and General Comments**

Reviewer #1: (No Response)

Reviewer #2: A nice paper but it could be improved by provision of more details regarding lipid identification.

PLOS authors have the option to publish the peer review history of their article (what does this mean?). If published, this will include your full peer review and any attached files.

Reviewer #1: No

Reviewer #2: No
---

## [Editor Report · Decision Letter 1]

5 Oct 2020

Dear Dr Wang,

We are pleased to inform you that your manuscript 'Quantitative lipidomic analysis of Ascaris suum' has been provisionally accepted for publication in PLOS Neglected Tropical Diseases.

Best regards,

Michael Cappello

Associate Editor

Sara Lustigman

Deputy Editor

---

## [Editor Report · Acceptance letter]

23 Oct 2020

Dear Prof. Gasser,

We are delighted to inform you that your manuscript, "Quantitative lipidomic analysis of *Ascaris suum*," has been formally accepted for publication in PLOS Neglected Tropical Diseases.

Best regards,

Shaden Kamhawi

co-Editor-in-Chief

Paul Brindley

co-Editor-in-Chief
